# Chlorhexidine cord care after a national scale-up as a newborn survival strategy: A survey in four regions of Ethiopia

**Ayalew Astatkie**[1]*, **Girma Mamo**[2], **Tilahun Bekele**[3], **Abdulaziz Adish**[2], **Sara Wuehler**[4], **Jennifer Busch-Hallen**[4], **Samson Gebremedhin**[5]

1 School of Public Health, College of Medicine and Health Sciences, Hawassa University, Hawassa, Ethiopia, 2 Nutrition International, Ethiopia Country Office, Addis Ababa, Ethiopia, 3 Center for Food Science and Nutrition, Addis Ababa University, Addis Ababa, Ethiopia, 4 Nutrition International, Head Quarter, Ottawa, Canada, 5 School of Public Health, Addis Ababa University, Addis Ababa, Ethiopia

* ayalewastatkie@gmail.com

## Abstract

### Introduction

Chlorhexidine cord care is an effective intervention to reduce neonatal infection and death in resource constrained settings. The Federal Ministry of Health of Ethiopia adopted chlorhexidine cord care in 2015, with national scale-up in 2017. However, there is lack of evidence on the provision of this important intervention in Ethiopia. In this paper, we report on the coverage and determinants of chlorhexidine cord care for newborns in Ethiopia.

### Methods

A standardized Nutrition International Monitoring System (NIMS) survey was conducted from January 01 to Feb 13, 2020 in four regions of Ethiopia (Tigray, Amhara, Oromia, and Southern Nations, Nationalities and Peoples Region [SNNPR]) on sample of 1020 women 0–11 months postpartum selected through a multistage cluster sampling approach. Data were collected using interviewer-administered questionnaires in the local languages through home-to-home visit. Accounting for the sampling design of the study, we analyzed the data using complex data analysis approach. Complex sample multivariable logistic regression was used to identify the determinants of chlorhexidine cord care practice.

### Results

Overall, chlorhexidine was reportedly applied to the umbilical cord at some point postpartum among 46.1% (95% confidence interval [CI]: 41.1%– 51.2%) of all newborns. Chlorhexidine cord care started within 24 hours after birth for 34.4% (95% CI: 29.5%– 39.6%) of newborns, though this varied widely across regions: from Oromia (24.4%) to Tigray (60.0%). Among the newborns who received chlorhexidine cord care, 48.3% received it for the recommended seven days or more. Further, neonates whose birth was assisted by skilled birth attendants had more than ten times higher odds of receiving chlorhexidine cord care, relative to those born without a skilled attendant (adjusted odds ratio [AOR]: 10.36, 95% CI: 3.73–28.75).

**Funding:** The study was funded by Nutrition International Ethiopia (https://www.nutritionintl. org/our-work/our-global-projects/africa/ethiopia/) and provided support in the form of consultancy fees to AA1, TB and SG. The specific roles of these authors are articulated in the 'author contributions' section. The funder had no role in study design, data collection and analysis, decision to publish, or preparation of the manuscript.

**Competing interests:** SG has been serving on the Editorial Board of PLoS ONE. GM, AA2, SW, and JBH work at Nutrition International (Ethiopia and Canada) that commissioned the survey. AA1, TB and SG received consultancy fees for conducting the survey. This does not alter our adherence to PLOS ONE policies on sharing data and materials. There are no patents, products in development or marketed products associated with this research to declare.

Besides, neonates born to mothers with knowledge of the benefit of chlorhexidine cord care had significantly higher odds of receiving chlorhexidine cord care relative to newborns born to mothers who did not have knowledge of the benefit of chlorhexidine cord care (AOR: 39.03, 95% CI: 21.45–71.04).

## Conclusion

A low proportion of newborns receive chlorhexidine cord care in Ethiopia. The practice of chlorhexidine cord care varies widely across regions and is limited mostly to births attended by skilled birth attendants. Efforts must continue to ensure women can reach skilled care at delivery, and to ensure adequate care for newborns who do not yet access skilled delivery.

## Introduction

Child health has been showing substantial improvement in Ethiopia over the past two decades. Under-five mortality has been reduced from its level of 123 per 1000 live births in 2005 to 55 per 1000 live births in 2019 [1]. As such, Ethiopia was able to attain the Millennium Development Goal (MDG) of reducing child mortality by two-thirds by 2015 three years ahead of the deadline. Ethiopia has shown the fastest annual rate (5.0%) of under-five mortality reduction in east Africa [2]. Ethiopia is also one of the ten low-income countries that were able to reduce under-five mortality by at least two-thirds between 1990 and 2018 [3]. Albeit the substantial improvements recorded over the years, child health still remains in peril in Ethiopia. With the under-five mortality in Ethiopia at 55 per 1000 live births [1], Ethiopia is one of the five countries in which half of all the global under-five deaths occur [3]. Accordingly, Ethiopia continues working towards further reduction of child mortality in line with the target set in the Sustainable Development Goal (SDG) of reducing child mortality to no more than 25 per 1000 live births by 2030 [4].

An important contributor to the high level of child mortality in Ethiopia is the slow decrease in neonatal mortality [2, 5], from 39 per 1000 live births only to 30 per 1000 live births between 2005 and 2019 [1]. Accordingly, Ethiopia has one of the highest neonatal mortality rates in the world [2], which accounts for 55% of all under-five deaths in the country [5]. Among the factors accounting for this high rate of neonatal mortality is neonatal infection entering through the cord [6, 7] due to unsafe birth and postnatal care practices. Wide spread application of various substances such as butter, petroleum jelly, and animal dung to the umbilical cord of newborn in Ethiopia [8, 9] may increase infection-risk of newborns. Neonatal infection has been consistently shown to be among the three leading causes of neonatal mortality in Ethiopia in both community-based [10–12] and hospital-based [13–15] studies with infection-specific mortalities ranging from 12% to 42%.

Although studies vary, most findings indicate that much of the death due to neonatal infections could be tackled by proper cord care using 7.1% chlorhexidine digluconate (hereafter referred to as "chlorhexidine"), while avoiding harmful cultural practices. Evidence from randomized controlled trials (RCTs) in three countries of south Asia has shown that chlorhexidine cord care significantly reduces both cord infection and all-cause neonatal mortality [7, 16, 17]. An RCT in Tanzania has reported that chlorhexidine cord care significantly reduced cord infection but had no significant effect on all-cause neonatal mortality [18], whereas an RCT from Zambia found no significant difference of both cord infection and all-cause neonatal mortality between neonates who received chlorhexidine cord care and dry cord care [19]. A

meta-analysis pooling results from the five RCTs (three from south Asia–Bangladesh, Nepal, and Pakistan; and the two from Sub-Saharan Africa–Tanzania and Zambia) found that chlorhexidine cord care resulted in 32% reduction of the risk of cord infection and 13% reduction of the risk of all cause mortality [20]. A recent review recommended chlorhexidine as a "low-cost, high-benefit intervention" to reduce mortality even among preterm and low-birth-weight neonates [21]. Considering the lack of hygienic conditions at health facilities and at home, early discharge from health facilities after birth, and possible application of harmful substances at home, Ethiopia has included chlorhexidine cord care in its "package of high-impact child survival interventions" for use both at a community level and health facilities since 2015 [22]. Subsequently, it was piloted in 2015 and 2016 and scaled up nationally in 2017.

However, chlorhexidine cord care doesn't seem to be a routine practice in Ethiopia. On the one hand, still about 50% of births take place at home [1], where chlorhexidine may not be readily available. On the other hand, even for births taking place in health facilities, chlorhexidine cord care is not universally practiced. A study in four zones of Ethiopia where chlorhexidine cord care introduction was piloted showed that 14 months after the introduction of the programme, 53% of neonates received chlorhexidine cord care [23]. The proportion of newborns born at home who received chlorhexidine cord care was reported to be only 5% [9]. Yet, there is lack of studies regarding the extent of practice of chlorhexidine cord care in Ethiopia following the national scale-up of chlorhexidine use for cord care. Studies documenting the determinants of chlorhexidine cord care are also lacking. The objective of the present paper, therefore, is to report on the coverage and determinants of chlorhexidine cord care based on a survey conducted in four regions of Ethiopia in which more than 85% of the national population resides.

## Materials and methods

### Study setting

The study was conducted in four regional states of Ethiopia, viz. Tigray, Amhara, Oromia, and Southern Nations, Nationalities and Peoples Region (SNNPR) (Fig 1). As per the 2007 National Housing Census of Ethiopia, these four regions cover more than 85% of the national population [24]. The study included 12 zones (second-level of the administrative division in Ethiopia) in the four regions–three from each region: Central Tigray, East Tigray and South Tigray zones of the Tigray region; West Gojam, East Gojam and Awi zones of Amhara region; Horo-Guduru, West Wollega and East Wollega zones of Oromia region; and Hadiya, Kembata-Tembaro and Sidama zones of the SNNPR region. A total of 71 districts (third-level administrative division) and 104 villages were covered in the 12 zones in which the study was conducted.

### Study design and population

This was a community-based cross-sectional survey conducted from January 01 to Feb 13, 2020 among women 0–11 months postpartum with live birth and permanently residing in the selected villages.

The survey was conducted as part of the baseline assessment for the Nutrition International's (NI's) Maternal, Newborn Health and Nutrition (MNHN) Programme planned to be implemented in the survey districts over 2020–2024. Promoting the use of Chlorhexidine for cord care is among the package of services included in the NI's MNHN programme.

### Sample size and sampling

As this study was part of a larger survey, sample size was estimated to ensure that it would be sufficient to estimate all key indicators covered by the NI-supported MNHN programme.

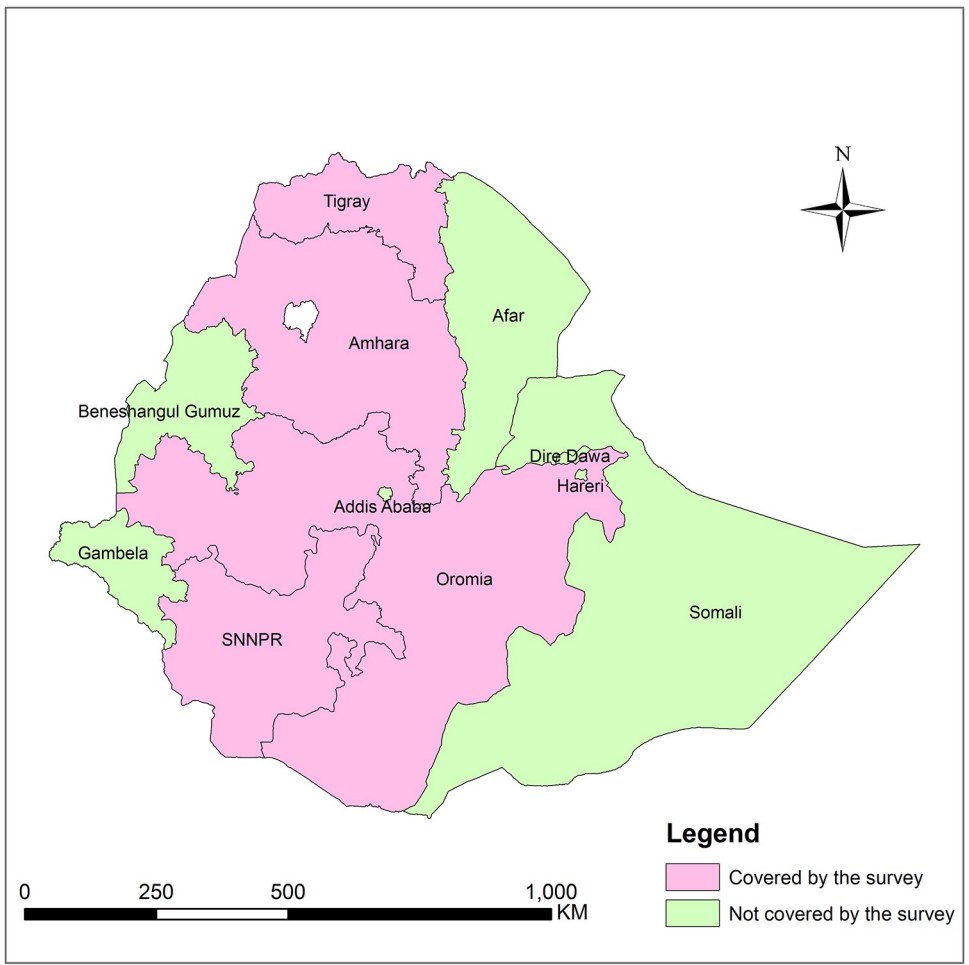

**Fig 1. Map of Ethiopia showing regions covered by the survey.** SNNPR: Southern Nations, Nationalities, and Peoples Region. Note: Currently there are two additional regions, namely Sidama and South West Ethiopia regions, which were created after the survey was conducted. The map is constructed based on shapefile obtained from open AFRICA (https://africaopendata.org/dataset/ethiopia-shapefiles; accessed on 01 February 2022).

Accordingly, a sample size of 1020 was found to be adequate for estimating all key variables. As for chlorhexidine in particular, assuming the expected proportion of chlorhexidine cord care among newborns to be 50% (to maximize the sample size) and considering a design effect of 2.0 to account for precision loss due to multistage cluster sampling, the sample size of 1020 used for the present study was sufficient to estimate the 95% confidence interval (CI) of the proportion of newborns who received chlorhexidine cord care within a margin of error of +/-4.34%. This sample size was also sufficient to analyze the determinants of chlorhexidine cord care. At the 95% confidence level and 80% power, taking the proportion of newborns who received chlorhexidine cord care in the reference categories of the determinants reported in this paper (region of residence, mother's education, skilled attendance at birth, and mother's knowledge of the benefit of chlorhexidine cord care) as the proportion of the outcome in the 'unexposed group', the sample size was adequate to detect statistically significant effects for adjusted odds ratios in the range of 1.7 to 3.0.

The multistage cluster sampling approach involved the following. In the first stage, 71 districts were identified using probability proportional to size (PPS) technique from across all 12

zones. For the second stage sampling, in most of the districts, one *kebele* (a lower administrative unit in Ethiopia) was selected at random. Because of relatively smaller population sizes in some of the districts, in a few districts (especially in Tigray region) PPS technique resulted in selection of 2–4 *kebeles* 2–4 *kebeles*. Ultimately, one village was selected from each *kebele* using simple random sampling (SRS) technique. In each selected village, a home-to-home search was done to identify eligible women. The first 10 eligible women 0–11 months postpartum with live birth identified during the home-to-home search were included in the study. If a sufficient number of eligible women was not found in a village, the remaining number of women was obtained from an adjacent village in the same *kebele*.

## Variables of the study

The dependent variable for this study was ***chlorhexidine cord care of neonates***. It was measured as the proportion of newborns who received chlorhexidine to the umbilical stump within 24 hours of birth in the previous one year. The information on chlorhexidine use for cord care was obtained through the mothers' self reports. A coloured picture of chlorhexidine tube was shown to the mothers to help them recall what chlorhexidine is.

Various plausible maternal sociodemographic, knowledge and health care-related variables were analyzed as the determinants of chlorhexidine core care. Sociodemographic variables included maternal education and region of residence. Maternal education was categorized as *no formal education*, *primary education*, *secondary education* and *higher education*, whereas region of residence comprised of the four regions in which the survey was conducted, namely *Tigray*, *Amhara*, *Oromia* and *SNNPR*. Maternal knowledge of the benefit of chlorhexidine cord care was another independent variable. Mothers who were able to name at least one of the three benefits of chlorhexidine cord care–namely '*it prevents infection*', '*it keeps the area clean*', and/or '*it prevents death*'–were categorized as having at least some knowledge (*yes*), and mothers who could not name any of those benefits were categorized as having no knowledge (*no*). Mothers' knowledge of how long to apply chlorhexidine to the umbilical stump was also included as an independent variable. Mothers who stated that chlorhexidine should be applied for seven or more days were considered as having knowledge (*yes*), while mothers who described the proper duration of chlorhexidine application to be less than seven days and those who couldn't describe chlorhexidine application as a way of caring for the cord were considered as having no knowledge of the correct duration of chlorhexidine cord care *(no)*. The health-care related variable was *skilled birth attendance during the last birth* (i.e., in the previous one year). Births attended by doctors/clinical officers and nurses/trained midwives were categorized as having been attended by skilled birth attendants (*yes*); otherwise, births were categorized as having not been attended by skilled birth attendants (*no*).

## Data collection tools and procedures

Data were gathered by 44 trained and experienced enumerators. Twenty-two supervisors (one supervisor per two data collectors) closely monitored and supervised the data collectors, and four regional coordinators oversaw the overall field operation. A team, composed of two data collectors and one supervisor, was responsible to collect data in approximately five clusters. Recruitment of the personnel was made based on multiple criteria including: educational status (at least diploma holders in health-related disciplines), local language proficiency, experience of work in similar surveys, acquaintance with electronic data collection applications and successful attainment of the skills applied during the survey's training.

Survey data collection was accomplished as per the Nutrition International Monitoring System (NIMS) survey toolkit and methods. NIMS includes a standardized set of tools: 1)

programme-specific indicators, related questionnaires with Open Data Kit (ODK) digital tools and pre-set indicator mapping; 2) quality control metrics during data collection to assess and quickly correct errors in plausibility and unlikely variability within and across data collectors; 3) and standard methods of assessing quality and reliability post-data collection to identify problems in attained sample size and implications on final data. NIMS tools include modules specific to the programmes supported by NI, such as the MNHN programme in Ethiopia, as well as a module on socio-economic indicators. The present analyses utilized data from both the socio-economic module and the MNHN module.

A structured questionnaire that was prepared in the English language and translated into the three official local languages of the four regions (Amharic, Afaan Oromoo and Tigrigna) was used for data collection. The questionnaire was converted into an electronic version using the ODK tool [25]. The ODK questionnaire incorporated all the four-language versions (English, Amharic, Afaan Oromoo and Tigrigna) so that the data collectors could easily switch between the languages as necessary. The ODK-format questionnaire was uploaded to the ODK Aggregate server on the Google cloud platform. Ultimately, data were gathered using android-based smart phones/tablets synchronized with the server. The questionnaire was administered to mothers 0–11 months postpartum through a home-to-home visit. The collected data were uploaded to the server by the data collectors on daily basis. The uploaded data were closely monitored by the research team throughout the survey implementation period.

Prior to the field deployment, the enumerators and supervisors received a 5-day training guided by a structured training manual. The training included description of the overall purpose of the study, explanation on the sampling approach of the study, basic principles of data collection, line-by-line discussion on the questionnaire, practicing filling the questionnaire on ODK smart phone/tablets, mock (role play) interviews, field practice and a review of relevant ethical practices of research involving human subjects.

## Statistical analysis

The dataset of the survey was downloaded from the ODK aggregate server and exported to the IBM SPSS Statistics for Windows, version 26 [IBM Corp., Armonk, NY, USA] for data cleaning, recoding and analysis. The data were analyzed taking account of the complex sampling design of the survey. For this purpose, a complex sample plan file was prepared in SPSS explicitly declaring the design variables–namely the stratum, cluster and weight variables. The stratum variable was the region of residence of the study participants. Accordingly, it comprised of four strata (i.e., the four regions in which the survey was conducted–Tigray, Amhara, Oromia, and SNNPR). The cluster variable was *kebele*. The analysis weight was calculated separately for each of the four strata as proportional contribution of each stratum to the total population in the in the study areas divided by proportional contribution of each stratum to the total sample size.

Complex sample descriptive analyses were conducted to obtain descriptive summaries such as proportions. The association of various plausible determinants with chlorhexidine cord care of neonates was investigated using complex sample logistic regression. The logistic regression commenced with a crude (bivariable) analysis. Variables with a p-value less than or equal to 0.25 on crude analysis and those deemed important were included into the multivariable model as per the suggestion in the literature [26, 27]. Accordingly, all five independent variables described above were included into the multivariable model. All variables except maternal education were selected based on the p-value cut-off of 0.25, while maternal education was included considering its importance as a determinant of use of a range of maternal and child health services. Subsequently, the model was re-fined and re-fitted by excluding variables not

significant on the multivariable analysis and whose removal does not substantially affect the overall fit of the model and the parameters of the individual variables remaining in the model. Thus, the final multivariable model contained four independent variables, namely *maternal education*, *region of residence*, *skilled attendance during the last birth*, and *maternal knowledge of the benefit of chlorhexidine cord care*.

The goodness of fit of the final model was evaluated using Wald test of model effects and pseudo R-squares. The Wald test of effect of the final model was significant (Wald F = 19.78; $p < 0.001$) and Nagalkerke's R-square was 0.58 both of which indicate a good model fit. Results of the association of chlorhexidine cord care with the different determinants were expressed in terms of adjusted odds ratios (AORs) with 95% CIs. The association was considered significant when the 95% CI of the AOR doesn't embrace the null value 1.

The data on which this manuscript is based is provided as a S1 Appendix.

## Ethical considerations

The study protocol was approved by the Institutional Review Board of the Ethiopian Public Health Association (EPHA) (EPHP/OG/4002/19). Data were collected after receiving oral informed consent from each of the survey participants. The collected information was kept strictly confidential and was made available only to the research team. Prior to the data collection, all personnel with full access to the survey data completed an internationally recognized training on protecting human research participants. Further, all data collectors received a one-day training on ethics of research involving human subjects.

## Results

### Sociodemographic characteristics

A total of 1020 mothers who gave birth in the 12 months preceding the survey were interviewed in 71 districts and 104 villages sampled from four regions of Ethiopia. About 96% of the mothers who took part in this survey were from households headed by males. While 95% of the respondents were wives of the household head, 2.3% of them were household heads. More than half (55.2%) of the mothers were in the 25–34 years age category. About 35% of the mothers had no formal education and 95.7% were married. Five hundred and ninety seven (58.6%) of the index children were less than six months old. Boys were slightly overrepresented with a male-to-female ratio of 1.07 (Table 1).

### Mothers' awareness of the benefit of chlorhexidine cord care

The present survey reveals considerable gap regarding the mothers' knowledge of cord care. Less than half (47.5%, 95% CI: 42.1%-53.0%) of the mothers were able to describe at least one benefit of or reason for applying chlorhexidine to the umbilical cord. While only 37.1% of the mothers knew that chlorhexidine prevents infection, 35.1% knew that chlorhexidine keeps the umbilical area clean. The proportion of mothers who could correctly point out that chlorhexidine should be applied to a newborn's cord stump for seven or more days was only 21.8% (95% CI: 18.80–25.1) (Table 2).

### Chlorhexidine cord care of neonates

About 46.1% (95% CI: 41.1%– 51.2%) of the newborns reportedly received chlorhexidine to the umbilical cord after birth. The proportion of newborns who received chlorhexidine to the umbilical cord stump within 24 hours of birth was 34.4% (95% CI: 29.5%– 39.6%). Less than

**Table 1. Basic characteristic of the study participants and index infants, four regions of Ethiopia, Feb 2020.**

| Variables (n = 1020) | Frequency | Percentage |
|---|---|---|
| **Region** (non-weighted sample size) | | |
| Oromia | 270 | 26.5 |
| Amhara | 250 | 24.5 |
| Tigray | 250 | 24.5 |
| SNNPR | 250 | 24.5 |
| **Educational status of the respondent** | | |
| No formal education | 356 | 34.9 |
| Primary | 420 | 41.2 |
| Secondary | 201 | 19.7 |
| Higher | 42 | 4.2 |
| **Marital status of the respondent** | | |
| Married/ cohabiting | 978 | 95.9 |
| Single | 21 | 2.0 |
| Divorced/separated | 20 | 2.0 |
| **Age of the respondent (years)** | | |
| 15–24 | 309 | 30.2 |
| 25–34 | 563 | 55.2 |
| 35 or above | 141 | 13.8 |
| Unknown | 8 | 0.8 |
| **Age of the child (months)** | | |
| 0–5 | 597 | 58.6 |
| 6–11 | 423 | 41.4 |
| **Sex of the child** | | |
| Male | 527 | 51.7 |
| Female | 493 | 48.3 |

half (48.3%) of the infants for whom chlorhexidine was applied to the umbilical cord received it for 7 days or more (Table 3).

## Determinants of chlorhexidine cord care of neonates

The highest rate (60%) of chlorhexidine use for cord care of neonates was found in Tigray region. Neonates born in Tigray region had about three-fold higher odds of receiving

**Table 2. Mothers' knowledge of chlorhexidine cord care, four regions of Ethiopia, Feb 2020.**

| Variables (n = 1020) | Frequency | Percentage |
|---|---|---|
| **Mather can name at least one benefit or reason to apply chlorhexidine to the umbilical cord at birth** | | |
| Yes | 485 | 47.5 |
| No | 535 | 52.5 |
| **Benefit of or reason for applying chlorhexidine at birth described by the mothers** | | |
| Prevents infection | 378 | 37.1 |
| Keeps the area clean | 358 | 35.1 |
| Prevents death | 150 | 14.7 |
| **Mother can describe how long to apply chlorhexidine to a newborn's cord stump ($\geq$ 7 days)** | | |
| Yes | 222 | 21.8 |
| No | 798 | 78.2 |

**Table 3. Chlorhexidine cord care of neonates, four regions of Ethiopia, Feb 2020.**

| Variables (n = 1020) | Frequency | Percentage |
|---|---|---|
| **Chlorhexidine applied to the umbilical cord after birth** | | |
| Yes | 471 | 46.1 |
| No | 521 | 51.1 |
| Not known | 28 | 2.8 |
| **Newborn received chlorhexidine to the umbilical cord stump within 24 hours of birth** | | |
| Yes | 351 | 34.4 |
| No | 669 | 65.6 |
| **Duration (number of days) chlorhexidine was applied (n = 470)** | | |
| Less than 7 days | 243 | 51.7 |
| 7 days or more | 227 | 48.3 |

chlorhexidine cord care relative to neonates born in SNNPR (AOR: 2.93: 95% CI: 1.54–5.57). About 41% of the neonates whose birth was assisted by skilled birth attendants received chlorhexidine cord care while only 4% of neonates whose birth was attended by non-skilled attendants received chlorhexidine cord care. As such, neonates whose birth was assisted by skilled birth attendants had more than ten-fold higher odds of receiving chlorhexidine cord care relative to neonates whose birth was not attended by skilled birth attendants (AOR: 10.36, 95% CI: 3.73–28.75). Further, neonates born to mothers who could name at least one benefit of chlorhexidine cord care had about 39-times higher odds of receiving chlorhexidine cord care compared to neonates born to mothers who cannot name any benefit of chlorhexidine cord care (Table 4).

**Table 4. Determinants of chlorhexidine cord care of neonates, four regions of Ethiopia, Feb 2020.**

| Independent variable | Newborn received chlorhexidine to the umbilical cord within 24 hours of birth | | Crude odds ratio (95% CI) | Adjusted odds ratio (95% CI) |
|---|---|---|---|---|
| | Yes, number (%) | No, number (%) | | |
| **Region of residence** | | | | |
| Tigray | 62 (60.0) | 41 (40.0) | **3.94 (2.15–7.20)** | **2.93 (1.54–5.57)** |
| Amahara | 136 (40.8) | 198 (59.2) | 1.81 (0.99–3.31) | 1.47 (0.76–2.84) |
| Oromia | 64 (24.4) | 199 (75.6) | 0.85 (0.44–1.53) | 2.03 (0.89–4.67) |
| SNNPR | 88 (27.6) | 231 (72.4) | 1 | 1 |
| **Mother's education** | | | | |
| No formal education | 127 (35.6) | 229 (64.4) | 1 | 1 |
| Primary education | 136 (32.3%) | 284 (67.7) | 0.87 (0.59–1.27) | 0.86 (0.53–1.38) |
| Secondary education | 71 (35.3) | 130 (64.7) | 0.99 (0.67–1.46) | 0.62 (0.36–1.04) |
| More than secondary | 17 (40.3) | 25 (59.7) | 1.22 (0.57–2.60) | 0.74 (0.24–2.26) |
| **Birth assisted by a skilled birth attendant** | | | | |
| Yes | 344 (41.2) | 491 (58.8) | **17.15 (7.47–39.35)** | **10.36 (3.73–28.75)** |
| No | 7 (3.9) | 179 (96.1) | 1 | 1 |
| **Mother can name at least one benefit or reason to apply chlorhexidine at birth** | | | | |
| Yes | 326 (67.2) | 159 (32.8) | **41.41 (21.49–79.78)** | **39.03 (21.45–71.04)** |
| No | 25 (4.7) | 510 (95.3) | 1 | 1 |

Note: Odds ratios in bold typeface indicate statistically significant effects. Wald test of model effects: Wald F = 19.78; p-value < 0.001. Nagelkerke R-square = 0.58.

CI, confidence interval; SNNPR, Southern Nations, Nationalities, and People's Region.

## Discussion

The present survey revealed that only about a third (34.4%) of neonates born in the preceding year received chlorhexidine cord care within 24 hours after birth. The extent of chlorhexidine cord care varied widely across regions with the highest rate (60%) in Tigray and the lowest (24.4%) in Oromia. Taking neonates born in the SNNPR as a reference, neonates born in Tigray had a 2.93 times higher odds of receiving chlorhexidine within 24 hours after birth. Chlorhexidine cord care practice also varied widely between births assisted by skilled birth attendants and births not assisted by skilled birth attendants. While 41.2% of neonates born assisted by skilled birth attendants received chlorhexidine cord care within 24 hours after birth, only 3.9% of neonates born without skilled birth attendance received chlorhexidine cord care within 24 hours postpartum. Accordingly, neonates born by skilled birth attendants have a 10.36 times higher odds of receiving chlorhexidine cord care within 24 hours postpartum relative to those born without skilled birth attendants. Mothers' knowledge of the benefit of chlorhexidine cord care was another important determinant of the practice of chlorhexidine cord care. Neonates born to mothers who can name at least one benefit of chlorhexidine cord care have about 39- times higher odds of receiving chlorhexidine cord care within 24 hours postpartum.

The fact that only about a third of newborns in Ethiopia receive chlorhexidine cord care within 24 hours after birth implies that most newborns (about two-third) remain at a higher risk of cord infection and subsequent sepsis and death. Even after taking account of newborns who receive chlorhexidine after 24 hours postpartum, still more than half of newborns (~54%) do not receive chlorhexidine cord care. This is a particular concern because cord cutting and tying practices which are not hygienic, particularly the practice of applying various local substances to the cord under unhygienic conditions [8], increase the risk of infection and its consequences. Inadequate supply and distribution of chlorhexidine may account for such lower rate of chlorhexidine cord care. Hence, the responsible health authorities need to make sure that there is adequate supply and equitable distribution of chlorhexidine if neonatal mortality due to sepsis is to be reduced.

The wide disparity in the practice of chlorhexidine cord care across the four regions may partly be attributed to variations in the proportions of births assisted by skilled birth attendants. As per the mini-Demographic and Health Survey (DHS) 2019 of Ethiopia [1], of the four regions (Tigray, Amhara, Oromia, and SNNPR), Tigray region has the highest rate (73.3%) of births assisted by skilled birth attendants, while Oromia has the lowest (43.7%). The proportions of newborns who got chlorhexidine application in the present study across the four regions followed a similar distribution as the proportion of births assisted by skilled birth attendants in mini-DHS 2019, which implies the rate of chlorhexidine cord care across the four regions to be directly proportional to the rate of skilled attendance at birth. As discussed below, our findings show that chlorhexidine cord care is provided almost exclusively for births assisted by skilled attendants. Hence, regions with higher rate of skilled birth attendance are likely to have higher rate of chlorhexidine cord care. Variations in the strength of the programme implementation and follow up, insufficient or inefficient supply and distribution of chlorhexidine and overall inequity in the distribution of health services across regions may also account for the observed disparity in chlorhexidine cord care.

As noted, the present study reveals that in Ethiopia chlorhexidine cord care is provided almost exclusively for neonates born assisted by skilled birth attendants. As per mini-DHS 2019 of Ethiopia [1], almost all births assisted by skilled birth attendants are health facility births except in Dire Dawa city. This means, chlorhexidine cord care is provided mainly for health facility births. Ethiopia's newborn and child survival strategy [22] recommends

chlorhexidine cord care for both community- and facility-level births. The World Health Organization (WHO) also recommends chlorhexidine cord care primarily for home births "in settings with high neonatal mortality" [28]. Yet, in addition to the low coverage, chlorhexidine cord care in Ethiopia is limited to health facility births contrary to the prevailing recommendations. Federal, regional and district health authorities need to ensure that chlorhexidine cord care be available and accessible to home births, which occur under unhygienic conditions predisposing the newborns to a high risk of infection and death.

Neonates born to mothers who have knowledge of the benefit of chlorhexidine cord care have higher odds of receiving chlorhexidine cord care within 24 hours after birth, as knowledgeable mothers may be more likely to remind or request the birth assistant to use chlorhexidine for cord care. Knowledgeable mothers may also prefer to give birth in a facility providing chlorhexidine cord care. However, this association could also be a reverse causality [29] whereby mothers whose newborn baby received chlorhexidine cord care may be provided with information on the benefits of chlorhexidine, and how and for how long to apply it, thus increasing the mothers' knowledge related to chlorhexidine.

The present study has limitations. First, it covered only four regions. Though the four regions addressed by the present study comprise more than 85% of the national population, the regions not included in the present study, especially, Afar, Benishangul-Gumuz, Gambella and Somali, are in a comparative disadvantage in terms of coverage of and access to health services. Thus, the coverage of chlorhexidine cord care of neonates is expected to be much lower in those regions. The result of the present study may therefore be an overestimate of the coverage of chlorhexidine cord care in Ethiopia. On the other hand, the present study focused mainly on the demand-side determinants. Supply-side determinants such as availability of sufficient chlorhexidine stock, and health workers' knowledge of and willingness to provide chlorhexidine cord care were not included. Thus, the present study has missed some important determinants that are amenable to interventions. Besides, relying on mother's recall of chlorhexidine use, especially in the first 24 hours can be problematic and the result might have been affected by recall bias. Further, only the first 10 women identified during the home-to-home search in each village were included in the study. Hence, the results could have been biased toward those women who live closest to where the search started.

In conclusion, the present survey revealed that the coverage of chlorhexidine cord care of newborns in the four regions, and likely all of Ethiopia is low. There is also a wide disparity in coverage across regions, and the care is provided mainly for births assisted by skilled birth attendants. There is a need to strengthen programmes on the use of chlorhexidine for cord care at all births across all regions. These efforts must continue to ensure women can reach skilled care at delivery, and to ensure adequate care for newborns who do not yet access skilled delivery.

## Supporting information

**S1 Appendix.**
(SAV)

## Acknowledgments

The authors are grateful to all study participants who willingly took part in the study. The data collectors and supervisors also deserve thanks.

## Author Contributions

**Conceptualization:** Ayalew Astatkie, Girma Mamo, Tilahun Bekele, Abdulaziz Adish, Sara Wuehler, Jennifer Busch-Hallen, Samson Gebremedhin.

**Data curation:** Ayalew Astatkie, Samson Gebremedhin.

**Formal analysis:** Ayalew Astatkie.

**Funding acquisition:** Ayalew Astatkie, Tilahun Bekele, Samson Gebremedhin.

**Investigation:** Ayalew Astatkie, Girma Mamo, Tilahun Bekele, Abdulaziz Adish, Sara Wuehler, Jennifer Busch-Hallen, Samson Gebremedhin.

**Methodology:** Ayalew Astatkie, Girma Mamo, Sara Wuehler, Jennifer Busch-Hallen, Samson Gebremedhin.

**Project administration:** Girma Mamo, Tilahun Bekele, Samson Gebremedhin.

**Resources:** Girma Mamo, Tilahun Bekele, Abdulaziz Adish, Samson Gebremedhin.

**Software:** Ayalew Astatkie, Samson Gebremedhin.

**Supervision:** Ayalew Astatkie, Girma Mamo, Tilahun Bekele, Abdulaziz Adish, Sara Wuehler, Jennifer Busch-Hallen, Samson Gebremedhin.

**Validation:** Ayalew Astatkie, Girma Mamo, Tilahun Bekele, Samson Gebremedhin.

**Visualization:** Ayalew Astatkie.

**Writing – original draft:** Ayalew Astatkie.

**Writing – review & editing:** Ayalew Astatkie, Girma Mamo, Tilahun Bekele, Abdulaziz Adish, Sara Wuehler, Jennifer Busch-Hallen, Samson Gebremedhin.

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
