## [Decision Letter · Decision Letter 0]

28 Jan 2022

PONE-D-21-40049Chlorhexidine cord care after a national scale-up as a newborn survival strategy: A survey in four regions of EthiopiaPLOS ONE

Dear Dr. Astatkie,

Thank you for submitting your manuscript to PLOS ONE. After careful consideration, we feel that it has merit but does not fully meet PLOS ONE’s publication criteria as it currently stands. Therefore, we invite you to submit a revised version of the manuscript that addresses the points raised during the review process.

We look forward to receiving your revised manuscript.

Kind regards,

Jai K Das

Academic Editor

PLOS ONE

Journal Requirements:

2 .Thank you for stating the following in the Acknowledgments Section of your manuscript: 

[The authors are grateful to all study participants who willingly took part in the study. The data collectors and supervisors also deserve thanks. The study was financially supported by Nutrition International Ethiopia.]

 [The study was funded by Nutrition International Ethiopia (https://www.nutritionintl.org/our-work/our-global-projects/africa/ethiopia/). The funder had no role in study design, data collection and analysis, decision to publish, or preparation of the manuscript.]

[GM, AA2, SW, and JBH work at Nutrition International (Ethiopia and Canada) that commissioned the survey. AA1, TB and SG received consultancy fees for conducting the survey. The authors declare that the conflict of interest declared did not alter their adherence to all PLOS one policies.]

Reviewers' comments:

Reviewer's Responses to Questions

**Comments to the Author**

1. Is the manuscript technically sound, and do the data support the conclusions?

Reviewer #1: Yes

Reviewer #2: Partly

2. Has the statistical analysis been performed appropriately and rigorously? 

Reviewer #1: Yes

Reviewer #2: No

3. Have the authors made all data underlying the findings in their manuscript fully available?

Reviewer #1: Yes

Reviewer #2: Yes

4. Is the manuscript presented in an intelligible fashion and written in standard English?

Reviewer #1: Yes

Reviewer #2: Yes

5. Review Comments to the Author

Reviewer #1: This study titled, "Chlorhexidine cord care after a national scale-up as a newborn survival strategy: A

survey in four regions of Ethiopia" assess the use of Chlorehexidine to umbilical cord care in 4 regions of Ethopia. The study is well designed and the methods are well described. The results are cleraly presented and the conclusions are appropriate. I have no major comments about the study.

Reviewer #2: The authors in this paper have attempted to asses the coverage and determinants of an evidence based intervention for reducing newborn morbidity and mortality in a high burden setting. However, I have the following major concerns with the study methods and rigor:

1. My main concern with the paper is that the objective of the study was not to asses the coverage and determinants of Chlorhexidine use after a national scale-up, but as the authors state in the paper that the survey was conducted as a part of the baseline assessment for the "Nutrition International’s (NI’s) Maternal, Newborn Health and Nutrition (MNHN) Programme planned to be implemented in the survey districts over 2020-2024. Promoting the use of Chlorhexidine for cord care is among the package of services included in the NI’s MNHN programme." Therefore, the variables assessed as determinants are the ones that were available from the survey findings rather than the potential determinants of the Chlorhexidine use. As a result, the association has only been determined with a handful of available variables (including maternal education, region of residence, skilled attendance during the last birth, and maternal knowledge of the benefit of chlorhexidine cord care). There are many other potential variables to be explored in the model that could impact the coverage including implementation coverage of national scale-up, healthcare provider, availability of chlorhexidine, healthcare worker knowledge etc. Without information of these critical variables, the findings stand incomplete.

2. My second concern with the study is self-reporting of chlorhexidine use. Considering that the sample included mothers 0-11 months post partum, I suspect that the findings are highly prone to reporting and recall bias.

My suggestion to the authors would be to re-write this paper as a baseline survey findings for the "Nutrition International’s (NI’s) Maternal, Newborn Health and Nutrition (MNHN) Programme" and report all the variables assessed rather than report as a study targeted to explore the coverage and determinants, which would be scientifically more rigorous and logical.

Some specific comments below:

Abstract:

In the abstract 'methods' section, the authors have provided the year when national scale-up was implemented. I would suggest that the authors also provide the year of the survey for context.

Introduction:

Correct line 65: "Ethiopia is one of the five countries in which half OF all the global under-five deaths occur"

Methods:

Under study setting, the authors quote "As per the 2007 National Housing Census of Ethiopia, these four regions cover more than 85% of the national population". I would suggest the authors replace this with updated figures, if available, as it seems too old.

Adding a geographical map in this section might help visualize the study settings for the readers more than the text.

Under 'sample size and sampling', the authors state that "As this study was part of a larger survey, sample size was estimated to ensure that it would be sufficient to estimate all key indicators covered by the NI-supported MNHN programme." Although the authors state that the sample size was sufficient to achieve the objectives of the present study and analyze the determinants of chlorhexidine cord care, I would suggest that the authors specify which variables have they considered as determinants for the sample size assumptions.

Correct line 199-200: "The present STUDY analyses utilized data from both the socio-economic module and the MNHN module."

Results:

The confidence intervals for the associations between chlorhexidine use and 'skilled birth attendance' and 'knowledge about benefit' is too wide signifying the points made earlier.

Discussion:

The authors have rightly pointed out in the discussion section that some of the variables found to be associated with the use of chlorhexidine are merely chance findings or a results of reverse causality for example skilled birth attendance and knowledge of benefits of chlorhexidine.

6. PLOS authors have the option to publish the peer review history of their article (what does this mean?). If published, this will include your full peer review and any attached files.

Reviewer #1: No

Reviewer #2: **Yes: **Rehana A Salam

---

## [Author Response · Author response to Decision Letter 0]

9 Feb 2022

Point-by-point response to reviewers’ comments

Reviewer #1: 

Comment: This study titled, "Chlorhexidine cord care after a national scale-up as a newborn survival strategy: A survey in four regions of Ethiopia" assess the use of Chlorehexidine to umbilical cord care in 4 regions of Ethiopia. The study is well designed and the methods are well described. The results are clearly presented and the conclusions are appropriate. I have no major comments about the study.

Response: Thank you for your kind and positive assessment of our manuscript.

Reviewer #2: 

Comment: The authors in this paper have attempted to assess the coverage and determinants of an evidence based intervention for reducing newborn morbidity and mortality in a high burden setting. However, I have the following major concerns with the study methods and rigor:

1. My main concern with the paper is that the objective of the study was not to assess the coverage and determinants of Chlorhexidine use after a national scale-up, but as the authors state in the paper that the survey was conducted as a part of the baseline assessment for the "Nutrition International’s (NI’s) Maternal, Newborn Health and Nutrition (MNHN) Programme planned to be implemented in the survey districts over 2020-2024. Promoting the use of Chlorhexidine for cord care is among the package of services included in the NI’s MNHN programme." Therefore, the variables assessed as determinants are the ones that were available from the survey findings rather than the potential determinants of the Chlorhexidine use. As a result, the association has only been determined with a handful of available variables (including maternal education, region of residence, skilled attendance during the last birth, and maternal knowledge of the benefit of chlorhexidine cord care). There are many other potential variables to be explored in the model that could impact the coverage including implementation coverage of national scale-up, healthcare provider, availability of chlorhexidine, healthcare worker knowledge etc. Without information of these critical variables, the findings stand incomplete.

Response: Thank you for raising this concern. As you correctly pointed out, the present work is part of a larger survey designed to address various maternal and newborn health and nutrition (MNHN) variables. The data on the use of chlorhexidine for cord care and its determinants was obtained from the mentioned larger survey. As pointed out, the data used lacks some important variables that could affect use of chlorhexidine for cord care. We have recognized this fact and addressed the issue in the discussion section of the paper. In the third paragraph of the discussion section (page 23, lines 359-362), we have argued that the variation in the coverage of chlorhexidine use for cord care across regions may be attributable to variations in the strength of the programme implementation and follow up. In that argument we are implying that even though the strength/intensity of the programme implementation was not measured and analyzed as a determinant, it could still be playing a role by operating through other variables such as region of residence. Further, in the sixth paragraph of the discussion section (pages 24-25, lines 389-393), we have discussed as a limitation of the study that it did not address all important determinants of chlorhexidine use for cord care. We especially noted that “supply-side determinants such as availability of sufficient chlorhexidine stock, and health workers’ knowledge of and willingness to provide chlorhexidine cord care were not included.” We feel that, given such information, readers can interpret the results accordingly. 

Comment: 2. My second concern with the study is self-reporting of chlorhexidine use. Considering that the sample included mothers 0-11 months post partum, I suspect that the findings are highly prone to reporting and recall bias.

Response: Thank you for this concern as well. We also have recognized the possibility of recall bias and have stated it as part of the limitation of the study in the sixth paragraph of the discussion section (pages 24-25, lines 393-395). Furthermore, in the Materials and Methods section, under Variables of the Study (page 9, lines 169-171), we have stated that “A coloured picture of chlorhexidine tube was shown to the mothers to help them recall what chlorhexidine is.” Therefore, both what has been done to minimize the possibility of recall bias and the limitation of the study vis-à-vis recall bias have been addressed in the paper. 

Comment: My suggestion to the authors would be to re-write this paper as a baseline survey findings for the "Nutrition International’s (NI’s) Maternal, Newborn Health and Nutrition (MNHN) Programme" and report all the variables assessed rather than report as a study targeted to explore the coverage and determinants, which would be scientifically more rigorous and logical.

Response: Thank you for the suggestion. The MNHN survey is a large survey addressing several maternal and newborn health and nutrition indicators. The results of that large survey couldn’t be condensed into a single journal article. Besides, of the various indicators addressed by the MNHN survey, a pronounced knowledge gap exists on the use of chlorhexidine for cord care and its determinants. We therefore aimed to contribute to the meagre literature pool on chlorhexidine use for cord care. Thus, this paper which focused on chlorhexidine use would be worthy in terms of conveying more needed information on an issue not sufficiently researched before. 

Comment: Some specific comments below:

Abstract:

In the abstract 'methods' section, the authors have provided the year when national scale-up was implemented. I would suggest that the authors also provide the year of the survey for context.

Response: Thank you for this comment. We now have included the survey period in the methods part of the abstract (page 2, line 29).

Comment: Introduction:

Correct line 65: "Ethiopia is one of the five countries in which half OF all the global under-five deaths occur"

Response: Thank you for pointing this out. Corrected accordingly (page 4, lines 64-65). 

Comment: Methods:

Under study setting, the authors quote "As per the 2007 National Housing Census of Ethiopia, these four regions cover more than 85% of the national population". I would suggest the authors replace this with updated figures, if available, as it seems too old.

Response: Thank you for this concern. The last census in Ethiopia was conducted in 2007. Any description of the population profile and population composition of Ethiopia is based on the 2007 census and projections emanating from it. We are therefore retaining the description which is based on the 2007 census as it was.

Comment: Adding a geographical map in this section might help visualize the study settings for the readers more than the text.

Response: Thank you for this suggestion. We now have included a map which distinguishes regions covered by the survey and those not covered by the survey. Please see Fig 1, which is also cited within text on page 7.

Comment: Under 'sample size and sampling', the authors state that "As this study was part of a larger survey, sample size was estimated to ensure that it would be sufficient to estimate all key indicators covered by the NI-supported MNHN programme." Although the authors state that the sample size was sufficient to achieve the objectives of the present study and analyze the determinants of chlorhexidine cord care, I would suggest that the authors specify which variables have they considered as determinants for the sample size assumptions.

Response: Thank you for this comment. We did a post-hoc analysis of the adequacy of the sample size to analyze determinants. Accordingly, we took the results on the determinants of chlorhexidine cord care reported in the presented paper and estimated the minimum detectable effect sizes in terms of odds ratios. We now have revised the description to make the message clearer (page 8-9, lines 148-154). 

Comment: Correct line 199-200: "The present STUDY analyses utilized data from both the socio-economic module and the MNHN module."

Response: Thank you for this concern. We found the original description to be more proper than the suggested one. Therefore, we retained the original sentence.

Comment: Results:

The confidence intervals for the associations between chlorhexidine use and 'skilled birth attendance' and 'knowledge about benefit' is too wide signifying the points made earlier.

Response: Yes, thank you for noting this. The 95% confidence intervals (CIs) of the adjusted odds ratios for 'skilled birth attendance' and 'knowledge about benefit' are very wide. Yet, looking even at the lower limit of the CIs, it is notable that the magnitude of association of those variables with the use of chlorhexidine core care is large.

Comment: Discussion:

The authors have rightly pointed out in the discussion section that some of the variables found to be associated with the use of chlorhexidine are merely chance findings or a results of reverse causality for example skilled birth attendance and knowledge of benefits of chlorhexidine.

Response: Thank you for noting this. Yes, we have provided possible and plausible explanations for the observed associations.

---

## [Decision Letter · Decision Letter 1]

26 Apr 2022

PONE-D-21-40049R1Chlorhexidine cord care after a national scale-up as a newborn survival strategy: A survey in four regions of EthiopiaPLOS ONE

Dear Dr. Astatkie,

Thank you for submitting your manuscript to PLOS ONE. After careful consideration, we feel that it has merit but does not fully meet PLOS ONE’s publication criteria as it currently stands. Therefore, we invite you to submit a revised version of the manuscript that addresses the points raised during the review process.

We look forward to receiving your revised manuscript.

Kind regards,

Jai K Das

Academic Editor

PLOS ONE

Journal Requirements:

Reviewers' comments:

Reviewer's Responses to Questions

**Comments to the Author**

1. If the authors have adequately addressed your comments raised in a previous round of review and you feel that this manuscript is now acceptable for publication, you may indicate that here to bypass the “Comments to the Author” section, enter your conflict of interest statement in the “Confidential to Editor” section, and submit your "Accept" recommendation.

Reviewer #2: All comments have been addressed

Reviewer #3: All comments have been addressed

2. Is the manuscript technically sound, and do the data support the conclusions?

Reviewer #2: (No Response)

Reviewer #3: Partly

3. Has the statistical analysis been performed appropriately and rigorously? 

Reviewer #2: (No Response)

Reviewer #3: Yes

4. Have the authors made all data underlying the findings in their manuscript fully available?

Reviewer #2: (No Response)

Reviewer #3: Yes

5. Is the manuscript presented in an intelligible fashion and written in standard English?

Reviewer #2: (No Response)

Reviewer #3: Yes

6. Review Comments to the Author

Reviewer #2: (No Response)

Reviewer #3: Though the authors have addressed all the comments, but I still believe the comment from Reviewer #2 Comment #1 has not been addressed sufficiently. I agree with reviewer that many important variables have been ignored and therefore the model stands incomplete. How do authors think these results should be interpreted when they also agree to this issue.

7. PLOS authors have the option to publish the peer review history of their article (what does this mean?). If published, this will include your full peer review and any attached files.

Reviewer #2: **Yes: **Rehana A Salam

Reviewer #3: No

---

## [Author Response · Author response to Decision Letter 1]

16 May 2022

Rebuttal letter (point-by-point response to journal requirements and reviewers’ comments)

Journal requirements 

Requirement: Please review your reference list to ensure that it is complete and correct. If you have cited papers that have been retracted, please include the rationale for doing so in the manuscript text, or remove these references and replace them with relevant current references. Any changes to the reference list should be mentioned in the rebuttal letter that accompanies your revised manuscript. If you need to cite a retracted article, indicate the article’s retracted status in the References list and also include a citation and full reference for the retraction notice.

Response: We have carefully reviewed all articles included in the reference list. All references are correct and complete. None of the articles in the reference list is a retracted one as of the date of writing this response.

Reviewer #3: 

Comment: Though the authors have addressed all the comments, but I still believe the comment from Reviewer #2 Comment #1 has not been addressed sufficiently. I agree with reviewer that many important variables have been ignored and therefore the model stands incomplete. How do authors think these results should be interpreted when they also agree to this issue

Response: Thank you for raising this point again. As you pointed out, this was a critical comment raised by Reviewer #2 during the first round of review. Accordingly, we have considered the comment duly, and provided an elaborate response with which Reviewer #2 agreed.

To reiterate our response to Reviewer #2’s comment during the first round of review, ‘… the present work is part of a larger survey designed to address various maternal and newborn health and nutrition (MNHN) variables. The data on the use of chlorhexidine for cord care and its determinants was obtained from the mentioned larger survey. As pointed out, the data used lacks some important variables that could affect use of chlorhexidine for cord care. We have recognized this fact and addressed the issue in the discussion section of the paper. In the third paragraph of the discussion section (page 23, lines 359-362), we have argued that the variation in the coverage of chlorhexidine use for cord care across regions may be attributable to variations in the strength of the programme implementation and follow up. In that argument we are implying that even though the strength/intensity of the programme implementation was not measured and analyzed as a determinant, it could still be playing a role by operating through other variables such as region of residence. Further, in the sixth paragraph of the discussion section (pages 24-25, lines 389-393), we have discussed as a limitation of the study that it did not address all important determinants of chlorhexidine use for cord care. We especially noted that “supply-side determinants such as availability of sufficient chlorhexidine stock, and health workers’ knowledge of and willingness to provide chlorhexidine cord care were not included.” We feel that, given such information, readers can interpret the results accordingly.’

As indicated in the response to Reviewer #2 during the first round of review, we have provided context in the discussion section vis-à-vis variables not addressed in the present study. We have elaborated the implications of and limitations related to variables not included in the study. Such information would enable readers to interpret the results accordingly. We hope that Reviewer #3 would also agree with this justification.

---

## [Decision Letter · Decision Letter 2]

4 Jul 2022

Chlorhexidine cord care after a national scale-up as a newborn survival strategy: A survey in four regions of Ethiopia

PONE-D-21-40049R2

Dear Dr. Astatkie,

We’re pleased to inform you that your manuscript has been judged scientifically suitable for publication and will be formally accepted for publication once it meets all outstanding technical requirements.

Kind regards,

Jai K Das

Academic Editor

PLOS ONE

---

## [Editor Report · Acceptance letter]

28 Jul 2022

PONE-D-21-40049R2 

Chlorhexidine cord care after a national scale-up as a newborn survival strategy: A survey in four regions of Ethiopia 

Dear Dr. Astatkie:

I'm pleased to inform you that your manuscript has been deemed suitable for publication in PLOS ONE. Congratulations! Your manuscript is now with our production department. 

Kind regards, 

on behalf of

Dr. Jai K Das 

Academic Editor

PLOS ONE